# TEMPORAL ABSTRACTIONS-AUGMENTED TEMPORALLY CONTRASTIVE LEARNING: AN ALTERNATIVE TO THE LAPLACIAN IN RL

## ABSTRACT

In reinforcement learning (RL), the graph Laplacian has proved to be a valuable tool in the task-agnostic setting, with applications ranging from option discovery to dynamics-aware metric learning. Conveniently, learning the Laplacian representation has recently been framed as the optimization of a temporally-contrastive objective to overcome its computational limitations in large or even continuous state spaces (Wu et al., 2019). However, this approach relies on a uniform access to the state space $\mathcal{S}$, and overlooks the exploration problem that emerges during the representation learning process. In this work, we reconcile such representation learning with exploration in a non-uniform prior setting, while recovering the expressive potential afforded by a uniform prior. Our approach leverages the learned representation to build a skill-based covering policy which in turn provides a better training distribution to extend and refine the representation. We also propose to integrate temporal abstractions captured by the learned skills into the representation, which encourages exploration and improves the representation's dynamics-awareness. We find that our method scales better to challenging environments, and that the learned skills can solve difficult continuous navigation tasks with sparse rewards, where standard skill discovery methods are limited.

## 1 INTRODUCTION

Representation learning has been at the core of many recent machine learning advances (c.f. Bengio et al., 2013). With the advent of deep reinforcement learning (Mnih et al., 2015), representation learning has also become one of the main topics of interest in reinforcement learning (RL). For example, in the goal-conditioned hierarchical setting (Vezhnevets et al., 2017; Nachum et al., 2019a), one learns a *representation* that maps observations to an abstract space, the representation space, in which the higher-level policy defines the desired behavior of the lower-level policy. Distance in the representation space can then be used to reward and guide the lower-level policy towards specific goal states. Moreover, environments with rich observations and complex dynamics (e.g., Bellemare et al., 2020) have motivated recent works on learning representations as controllable or contingent features (Bengio et al., 2017; Choi et al., 2019), on top of which one can potentially learn latent models in the perspective of planning (Hafner et al., 2019b; Nasiriany et al., 2019; Schrittwieser et al., 2020) and control (Watter et al., 2015; Banijamali et al., 2018; Hafner et al., 2019a).

In this work, we are interested in the reward-agnostic setting in which an RL agent first interacts with the environment to build a representation, $\phi$, of the state space, $\mathcal{S}$, without relying on any task-specific reward signal. This representation can later be used to solve a task posed in the environment in the form of a reward function. In this setting, the environment dynamics are the only informative interaction channel available to the agent. This has naturally motivated graph Laplacian-based methods to address the task-agnostic phase; where the graph vertices correspond to the states and its edges to the transitions probabilities. The Laplacian's eigenvectors can been leveraged as a holistic state representation, termed the Laplacian representation, which captures the environment's dynamics structure and geometry (Mahadevan, 2005; Mahadevan & Maggioni, 2007).

Wu et al. (2019) recently proposed an efficient approximation of the Laplacian representation (LAP-REP) by framing the graph drawing objective as a temporally-contrastive loss (see Section 2.2).

While this formulation works around potentially prohibitive eigendecompositions, which extends the representation's applicability to large and continuous state spaces, it assumes access to a uniform sampling prior over $\mathcal{S}$. In practice, this translates in the ability to reset the agent to a *uniform* random starting state in the environment, which artificially alleviates the exploration problem. As we will show in Section 4, the uniformity of that prior is crucial for the quality of the learned representation. However, such sampling is not trivial in the absence of the uniform prior privilege, since the agent has to learn to explore the state space to be able to access arbitrary states. In effect, one must handle the exploration along the representation learning in order to preserve the representation's quality. In this work, we propose a representation learning framework that conciliate a similar temporally-contrastive approach with exploration in the task-agnostic setting.

In practice, the representation is trained on data collected with a uniformly random policy, $\pi_\mu$ (random walk trajectories). Without a uniform access to the state space, the collected data is concentrated around accessible starting states. To achieve a better data collection, we tie the representation learning problem to that of learning a covering strategy. Briefly, our method consists in using the available representation to learn a skill-based (hierarchical) covering policy that is in turn used to discover yet unseen parts of the state space, providing novel data to refine and expand the representation. Our approach, illustrated in Figure 1, is inspired by the cyclic option discovery framework (Machado, 2019), which motivated several related methods (Machado et al., 2017; 2018; Jinnai et al., 2020). In addition, we propose to integrate the temporal abstractions learned by the skills in the contrastive representation learning objective to encourage temporally-extended exploration and enforce the representation's *dynamics-awareness*, i.e. how representative the $\phi$-induced euclidean distance is of distances in the state space.

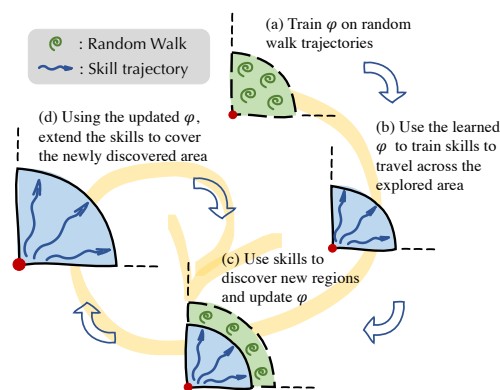

Figure 1: Our representation is trained to encode the area that the agent has learned to cover. Skills are continuously trained on the representation to discover new areas where novel data is collected to refine the representation, progressively extending its coverage. Similar incremental discovery is at the core of several works (Ecoffet et al., 2021; Pong et al., 2019; Machado, 2019).

We empirically show our agent's ability to progressively explore the state space and to consistently extend the representation covered domain in a non-uniform prior setting. We show that our representation leads to better value predictions than LAP-REP, and that it recovers the representation quality expected from a uniform prior. We also evaluate our representation in shaping rewards for goal-achieving tasks, and we show it outperforms LAP-REP, confirming both its superior ability in capturing dynamics and in scaling to *larger* environments. Finally, the skills learned in our framework also prove to be successful at difficult continuous navigation tasks with **sparse** rewards, where other standard skill discovery methods are limited.

## 2 PRELIMINARIES

### 2.1 TASK-AGNOSTIC REINFORCEMENT LEARNING

We describe a task-agnostic RL environment as a task-agnostic Markov decision process (MDP) $\mathcal{M} = (\mathcal{S}, \mathcal{A}, P, \gamma, d_0)$ where $\mathcal{S}$ is the state space, $\mathcal{A}$ the action space, $P : \mathcal{S} \times \mathcal{A} \rightarrow \Delta(\mathcal{S})$ the transition dynamics defining the next state distribution given current state and action taken, $\gamma \in [0, 1)$ the discount factor, and $d_0$ the initial state distribution. A policy $\pi : \mathcal{S} \rightarrow \Delta(\mathcal{A})$ maps states $s \in \mathcal{S}$ to distributions over actions.

Knowledge acquired from task-agnostic interactions with the environment (e.g., a representation or a policy) can then be leveraged for specific tasks. A task is instantiated with a reward function, $R : \mathcal{S} \rightarrow \mathbb{R}$, which is combined with the task-agnostic MDP. The task objective is to find the

optimal policy maximizing the expected discounted return, $\mathbb{E}_{\pi,d_0}\left[\sum_t \gamma^t R(s_t, a_t)\right]$, starting from state $s_0 \sim d_0$ and acting according to $a_t \sim \pi(\cdot|s_t)$.

## 2.2 THE LAPLACIAN REPRESENTATION

The Laplacian representation (LAP-REP), as proposed by Wu et al. (2019), can be learned with the following contrastive objective:

$$\mathcal{L}_{Lap}(\phi; \mathcal{D}_{\pi_\mu}) = \underset{(u,v) \sim \mathcal{D}_{\pi_\mu}}{\mathbb{E}}\left[\|\phi(u) - \phi(v)\|_2^2\right] + \beta \underset{\substack{u \sim \mathcal{D}_{\pi_\mu} \\ v \sim \mathcal{D}_{\pi_\mu}}}{\mathbb{E}}\left[(\phi(u)^\top \phi(v))^2 - \|\phi(u)\|_2^2 - \|\phi(v)\|_2^2\right], \quad (1)$$

where $\beta$ is a hyperparameter, $\pi_\mu$ is the uniformly random policy, $\mathcal{D}_{\pi_\mu}$ a set of trajectories from $\pi_\mu$ (random walks). We use $(u, v) \sim \mathcal{D}_{\pi_\mu}$ to denote the sampling of a random transition from $\mathcal{D}_{\pi_\mu}$, and similarly $u \sim \mathcal{D}_{\pi_\mu}$ for a random state. Wu et al. (2019) showed the competitiveness of the Laplacian representation when provided with a uniform prior over $\mathcal{S}$ during the collection of $\mathcal{D}_{\pi_\mu}$. Their objective (Eq. 1) is a *temporally-contrastive* loss: it is comprised of an attractive term that forces temporally close states to have similar representations and a repulsive term that keeps temporally far states' representations far apart. Here, the repulsive term was specifically derived from the orthonormality constraint of the Laplacian eigenvectors.

## 2.3 THE NON-UNIFORM PRIOR SETTING

In RL, representation learning is deeply coupled to the problem of exploration. Indeed, the induced state distribution defines the representation's training distribution. However, LAP-REP (Wu et al., 2019) has been learned in the specific *uniform prior* setting that alleviates the exploration challenge. In this setting, $\mathcal{D}_{\pi_\mu}$, from Eq. 1, is a collection of random walks with uniformly random starting states, which provides a uniform training distribution to the representation learning objective. In the case of a non-uniform prior, the induced visitation distribution can be quite concentrated around the start state distribution when solely relying on random walks, hence the need for an exploration strategy for a better covering distribution.

To study the problem described above, we investigate the setting in which the environment has a fixed predefined state $s_0$ to which it resets with a probability $p_r$ every $K$ steps; with $K$ of the order of diameter of $\mathcal{S}$. With a uniformly random behavior policy, this setting is equivalent to a initial state distribution that is concentrated around $s_0$ and whose density decays exponentially away from it. We will refer to this setting as the *non-uniform prior* (non-$\mu$) setting, as opposed to the *uniform prior* ($\mu$) setting where the agent has access to the uniform state distribution.

## 3 TEMPORAL ABSTRACTIONS-AUGMENTED REPRESENTATION LEARNING

In this section, we present Temporal Abstractions-augmented Temporally-Contrastive learning (TATC), a representation learning approach in which the representation works in tandem with a skill-based covering policy for a better representation learning in the non-uniform prior setting. We first propose an alternative objective to Eq. 1 that suits this setting, then describe the exploratory policy training. Finally, we introduce an augmentation of the proposed objective based on the learned temporal abstractions to improve exploration and enforce the representation's dynamics-awareness.

## 3.1 TEMPORALLY-CONTRASTIVE REPRESENTATION OBJECTIVE

As mentioned in Section 2.2, the repulsive term in LAP-REP's objective (Eq. 1) derives from the eigenvectors' orthonormality constraint. However, because the environment is expected to be progressively covered in the non-uniform prior setting, the orthonormality constraint can make online representation learning highly non-stationary.[1] For this reason, we adopt the following objective with a generic repulsive term that is more amenable to online learning:

$$\mathcal{L}_{cont}(\phi; \mathcal{D}_{\pi_\mu}) := \underset{(u,v) \sim \mathcal{D}_{\pi_\mu}}{\mathbb{E}}\left[\|\phi(u) - \phi(v)\|_2^2\right] + \beta \underset{\substack{u \sim \mathcal{D}_{\pi_\mu} \\ v \sim \mathcal{D}_{\pi_\mu}}}{\mathbb{E}}\left[\exp(-\|\phi(u) - \phi(v)\|_2)\right]. \quad (2)$$

---

[1]In general, even within a given matrix's perturbation neighborhood, its eigenvectors can show a highly nonlinear sensitivity (Trefethen & Bau, 1997).

## 3.2 Representation-based Covering Policy

In the non-uniform prior setting, exploration is required to provide the representation with a better training distribution. To this purpose, we adopt a hierarchical RL approach to leverage the exploratory efficiency of options (Sutton et al., 1999; Nachum et al., 2019b), or skills. The agent acts according to a bi-level policy $(\pi_{\text{hi}}, \pi_{\text{low}})$. The high-level policy $\pi_{\text{hi}} : \mathcal{S} \to \Delta(\Omega)$ defines, at each state $s$, a distribution over a set $\Omega$ of directions (unit vectors) in the representation space $(\Omega = \{\boldsymbol{\delta} \mid \boldsymbol{\delta} \in \mathbb{R}^d, \|\boldsymbol{\delta}\|_2 = 1\})$. Each direction corresponds to a fixed length skill encoded by the low-level policy $\pi_{\text{low}} : \mathcal{S} \times \Omega \to \Delta(\mathcal{A})$. These skills are expected to travel *in the representation space* along the directions instructed by $\pi_{\text{hi}}$. In short, given a sampled direction $\pi_{\text{hi}}(\cdot|s) \sim \boldsymbol{\delta} \in \Omega$, the low-level policy executes the directional skill $\pi_{\text{low}}(\cdot|s, \boldsymbol{\delta})$ for a fixed number of steps $c$ before a new direction is sampled.

Now, we describe the intrinsic rewards used to train the policies $\pi_{\text{hi}}$ and $\pi_{\text{low}}$.

**Low-level Policy.** $\pi_{\text{low}}$ is simply trained to follow directions defined by $\pi_{\text{hi}}$ in the representation space. For a given $\boldsymbol{\delta} \in \Omega \subset \mathbb{R}^d$, the corresponding skill $\pi_{\text{low}}(\cdot|s, \boldsymbol{\delta})$ is trained to maximize the intrinsic reward function:

$$r^{\boldsymbol{\delta}}(s, s') \coloneqq \cos(\boldsymbol{\delta}, \phi(s') - \phi(s)) = \frac{\boldsymbol{\delta}^\top (\phi(s') - \phi(s))}{\|\phi(s') - \phi(s)\|} \tag{3}$$

where $(s, s')$ is an observed state transition, and $\phi$ the representation being learned. We use the cosine similarity as a way to encourage learning diverse directional skills. Indeed, skills co-specialization is avoided by rewarding the agent for the steps induced along the instructed direction $\boldsymbol{\delta}$ regardless of their magnitudes.

**High-level Policy.** The high-level policy is expected to guide the covering strategy. It should do so by sampling the skills of the most promising directions in terms of exploration: affording new discoveries while avoiding to spend more time than needed in previously explored areas. For this purpose, we design a reward function defined over a sequence of $L$ consecutive skills. Let $\{s_k^{\text{hi}}\}_{k=1}^L$ be the sequence of their initial states and their respective sampled directions $\boldsymbol{\delta}_k \sim \pi_{\text{hi}}(\cdot|s_k^{\text{hi}})$. Since $\phi$ is trained to capture the dynamics, the travelled distance in $\phi$'s space is a good proxy of how far the choices made by $\pi_{\text{hi}}$ eventually brought the agent in the environment. Therefore, for a given high-level trajectory, $\tau^{\text{hi}} = (s_1^{\text{hi}}, s_2^{\text{hi}}, ..., s_L^{\text{hi}}, s_f^{\text{hi}})$, with $s_f^{\text{hi}}$ the final state reached by the last skill, the high-level policy is trained to maximize the following quantities:

$$\forall k \in \{1, ..., L\}, R^{\text{hi}}(s_k^{\text{hi}}, \boldsymbol{\delta}_k) \coloneqq \|\phi(s_1^{\text{hi}}) - \phi(s_f^{\text{hi}})\|_2, \tag{4}$$

where $\boldsymbol{\delta}_k \sim \pi_{\text{hi}}(\cdot|s_k^{\text{hi}})$ is the direction sampled at $s_k^{\text{hi}}$. From the policy optimization perspective, each of these quantities plays the role of the return cumulated along the sampled high-level trajectory and *not* just a single (high-level) step reward. This term looks at reaching $s_f^{\text{hi}}$ as the result of a sequential collaboration of $L$ skills, rewarding them equally. It values how far this sequence of skills has eventually brought the agent.

These policy training choices are closely related to how the representation is trained. Indeed, the exploratory behavior emerges from the interaction between the policy and the representation while training. In the following section, we describe how the representation benefits from the temporal abstractions learned by the covering policy $(\pi_{\text{hi}}, \pi_{\text{low}})$.

## 3.3 Augmenting Representation Learning with Temporal Abstractions

A skill abstracts a temporally-extended and structured behavior in a single action. As a *temporal abstraction*, it represents a factorized knowledge of the environment dynamics in the form of a policy (the directional skill policy). Here, we propose to leverage these temporal abstractions in the representation objective in the favor of better capturing the environment dynamics. In order to preserve the temporal contrast of the base objective (Eq. 2), we augment it with the following contracting term along the skills trajectories:

$$\mathcal{B}(\phi; \mathcal{D}_s) \coloneqq \mathbb{E}_{\substack{\tau_{\boldsymbol{\delta}} \sim \mathcal{D}_s \\ \tau_{\boldsymbol{\delta}} = (s_0, ..., s_c)}} \left[ \sum_{k=0}^{c-1} \|\phi(s_k) - \phi(s_{k+1})\| \right], \tag{5}$$

where $\mathcal{D}_s$ is a set of collected skills trajectories. By minimizing this term, $\phi$ integrates temporally-extended dynamics: areas connected by skills are brought closer in the representation space. This term will be referred to as the *boredom* term due to its exploratory virtue, explained in the following.

**How does boredom help exploration ?**  The interaction of the high-level policy reward function (Eq. 4) and this boredom term (Eq. 5) induces a progressive exploration mechanism. In effect, $\pi_{\text{hi}}$ samples skills that travel further more often, i.e. with larger $R^{\text{hi}}$. The more often a skill is sampled, the less rewarding it becomes due the minimization of $\mathcal{B}(\phi)$. This will increase the probability of sampling the remaining under-sampled skills, hence encouraging the exploration of less visited parts of the state space. In short, the interplay between the policy and the representation dynamically fights what can be considered as accumulated *boredom* along over-sampled skills trajectories which increases the agent curiosity and urge it to explore.

Finally, the proposed objective to train the representation $\phi$ consists in the objective in Eq. 2 augmented with the boredom term (Eq. 5), and can be written as

$$\mathcal{L}_{\text{TATC}}(\phi; \mathcal{D}_s, \mathcal{D}_{\pi_\mu}) := \mathcal{L}_{cont}(\phi; \mathcal{D}_{\pi_\mu}) + \beta' \mathcal{B}(\phi; \mathcal{D}_s) \tag{6}$$

with $\beta'$ a hyperparameter controlling the strength of boredom term. A detailed description of TATC training is available in the Appendix A (Algorithm 1).

## 4 EXPERIMENTS

In this section, we investigate the behavior of TATC in two types of environments: gridworld environments with discrete state and action spaces, and a continuous navigation environment (MuJoCo, Todorov et al. (2012)) for continuous state and action spaces. Implementation details of all the experiments in this section can be found in Appendix C.

### 4.1 GRIDWORLD

For gridworld environments, we evaluate our approach in three different domains: U-MAZE, T-MAZE and 4-ROOMS. These environments, visualized in Figure 2, raise different explorations challenges. U-MAZE is the simplest but the most relevant environment to test the dynamics-awareness of the representations[2]; T-MAZE raises the challenge of splitting the exploration focus at an intersection while maintaining exploration and coverage in both corridors; 4-ROOMS is similar to U-MAZE, but requires learning more controlled skills to efficiently move from one room to another.

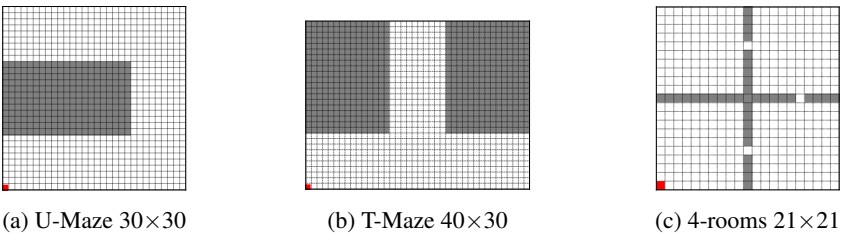

(a) U-Maze 30×30          (b) T-Maze 40×30          (c) 4-rooms 21×21

Figure 2: The gridworld domains with the fixed initial state $s_0$ highlighted in red.

The states are one-hot encoded such that no positional information is provided to the agent. For our method, we learn a 2D representation ($d = 2$), and define $\Omega$ as a set of 8 unit vectors equally spaced on the unit sphere (see Appendix C).

### 4.1.1 PROGRESSIVE REPRESENTATION LEARNING

Figure 3 shows the progression of the representations throughout training. The agent progressively explores the environment starting from $s_0$, builds the representation by continuously integrating newly discovered parts.

---

[2]The presence of the wall makes L2-distance in xy-coordinates deceptive. The L2-distance in a dynamics-aware representation space should correct for that.

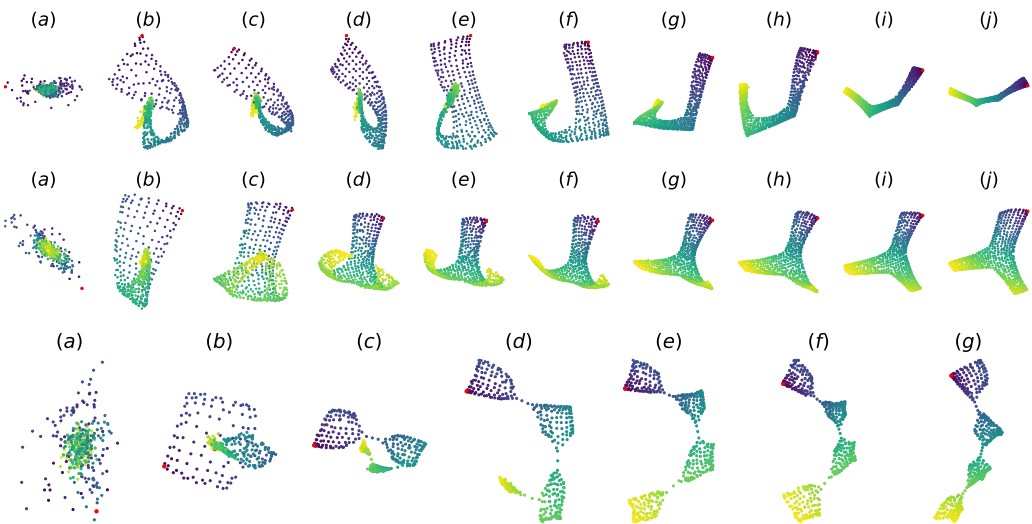

Figure 3: TATC representations learned throughout the training. Axes scales were equalized. Top row (**R1**): U-MAZE. Middle row (**R2**): T-MAZE. Bottom row (**R3**): 4-ROOMS. The colors reflect the distances in terms of the dynamics. They can be seen as quantities proportional to the length of the shortest path from $s_0$ (marked in red) to the represented state.

**U-MAZE.** The agent starts from the bottom left corner of the maze. Figure 3 shows how the representation progressively expands until reaching the first corner (Fig.3-R1,a-e). During this phase, the agent learns skills to travel further away from $s_0$ along the corridor. At this stage, the rest of the environment is still not explored. The remaining phase (Fig.3-R1,f-j) shows not only the complete discovery of the corridor but also the flattening of the full domain representation: placing the last corner further from the starting corner than the intermediate ones indicates the representation's success in capturing the maze dynamics.

**T-MAZE.** The agent starts from the bottom left corner of the maze. As in the U-Maze, it starts learning to travel along the corridor (Fig.3-R2,a-b) until reaching the intersection. There, the exploration focus is shared between both possible paths whose representations are progressively disentangled (Fig.3-R2,c-f). Eventually, the agent fully explores both corridors and finalizes its representation. Note that, the discovery of one of the corridors did not hinder finishing the discovery of the other. The boredom term proved to be important for such property (see Appendix B).

**4-ROOMS.** The agents starts in the first room. It progressively discovers and learns about other rooms. Once the domain is fully explored, and similarly to U-MAZE, the representation straightens, reflecting a holistic understanding of the environment dynamics.

**Boredom Ablation Study.** Appendix B provides an ablation study showing the importance of the boredom term for the agent's exploratory behavior and the representation's dynamics-awareness.

### 4.1.2   EVALUATING THE LEARNED REPRESENTATION

We now compare our representation against LAP-REP (Wu et al., 2019) in the non-uniform prior setting. First, to appreciate the sensitivity of LAP-REP to the uniformity of said prior, we trained LAP-REP in two settings: (i) the uniform prior setting where the agent can be set to any arbitrary state, as done by Wu et al. (2019), (ii) the non-uniform prior setting defined in Section 3. We show that LAP-REP is quite sensitive to this change in distribution while TATC recovers the expressive potential of LAP-REP learned with a uniform prior.

**Prediction.** To evaluate the learned representations, we first consider how well they linearly approximate a given task's optimal value function. To do so, we train an actor-critic agent (Mnih et al., 2016) with a linear critic on top of each representation. In Figure 4, we note a significant loss in the expressive power of LAP-REP when the access to the state space is not uniformly distributed anymore. This figure also shows that TATC outperforms LAP-REP in the non-uniform prior setting, and

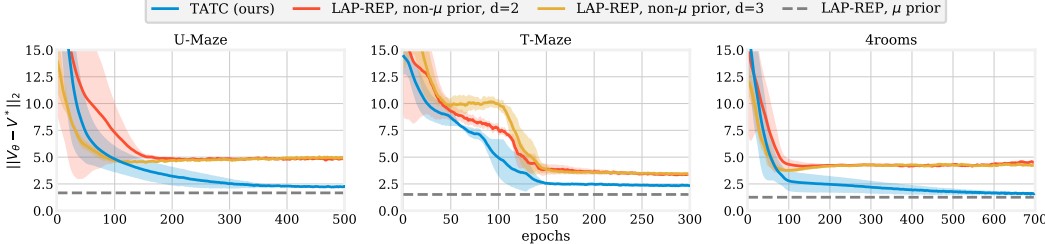

Figure 4: Learned representation's ability to approximate the value function. LAP-REP was learned in the same non-uniform prior setting (non-$\mu$) with $d = 2$ and $d = 3$ (no improvement was observed for higher values). The dashed line gives the performance of LAP-REP in the uniform prior setting ($\mu$). TATC outperforms LAP-REP in non-$\mu$ setting, and succeeds in recovering its expressive power when learned from the uniform prior. Performances were averaged over 5 different runs.

succeeds in recovering LAP-REP's expressive power when it is learned with the unrealistic uniform prior.

**Control.** We also compare the representations from the perspective of control, by training a deep actor-critic agent on top of each representation to solve a goal-reaching task in the same domains as above. The agent is only rewarded upon reaching the goal state ($r = 1$). Figure 5 shows that TATC consistently outperforms LAP-REP, which confirms the competitive quality of our representation.

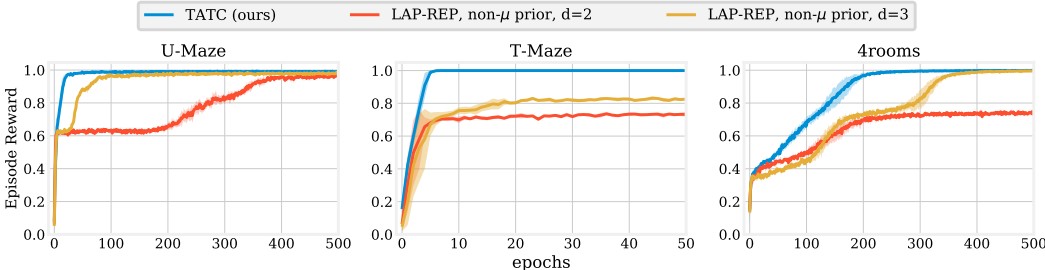

Figure 5: Control performance (episode reward) in the fixed initial state setting (non-uniform prior). Performances were averaged over 5 different runs.

### 4.2 CONTINUOUS CONTROL

The second set of experiments, which focuses on continuous state and action spaces, is conducted on AntMaze which is essentially a MuJoCo counterpart of U-MAZE where a four-legged agent has to learn to control its joints to maneuver along a U-shaped corridor.

To visualize our learned representation in this environment, Figure 6 depicts a grid of positional states in the environment domain and their mapped representations. Similarly to U-MAZE, the learned representation translates the environment dynamics by placing the end of the corridor (top left) away from the initial state (represented in red) than the intermediate corners (top and bottom right).

#### 4.2.1 REWARD SHAPING WITH LEARNED REPRESENTATION

We first demonstrate how our learned representation is able to improve an RL agent's performance when the distances in the representation space are used for reward shaping, the same setting in which Wu et al. (2019) evaluated LAP-REP. We define a goal-achieving task by setting a goal state $g$ at the end of the corridor (top left). The objective is to learn to navigate to a state $s$ close enough to the goal area ($\|s - g\|_2 \leq \epsilon$). We define the reward function based on the distance in representation space (TATC and LAP-REP). More specifically, we train a soft actor-critic (SAC) agent (Haarnoja et al., 2018) to reach the goal with a **dense** reward defined as $r_t^{dense} = -\|\phi(s_{t+1}) - \phi(g)\|_2$. Similarly

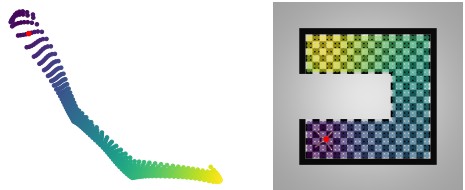 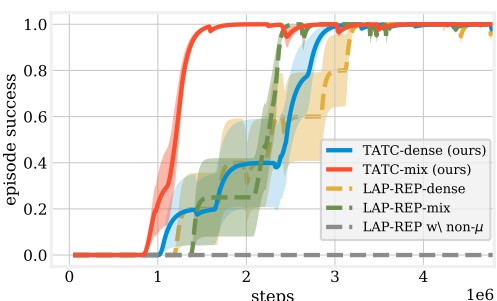

Figure 6: TATC learned representation visualized on a grid of positional states. Colors reflect the distance in the representation space from the initial state, highlighted in red. Axes scales were equalized. We can visually appreciate how the U-shaped continuous state domain is mapped to a flatter manifold reflecting the presence of the wall.

Figure 7: Results of reward shaping using learned representations: Performances were averaged over 5 different runs and then exponentially smoothed (0.9) for better visualization.

to Wu et al. (2019), we also compare against the half-half **mix** of the dense reward and the sparse reward $r_t^{mix} = 0.5 \cdot r_t^{dense} + 0.5 \cdot \mathbb{1}\left[\|s_{t+1} - g\|_2 \leq \epsilon\right]$.

For this evaluation, we used a larger environment than those used by Wu et al. (2019), making it a more challenging task. Unlike our representation, LAP-REP was learned with a *uniform* prior over $S$ as in Wu et al. (2019), but with $d = 2$ instead of $d = 20$. Figure 7 shows that our representation is effective in reward shaping, with both **mix** and **dense** variants , and enjoys a comparable if not superior dynamics-awareness to LAP-REP, even when TATC is learned from a non-uniform prior and LAP-REP is learned from a uniform one. Note that LAP-REP with a non-uniform prior is unable to guide the agent to success.

This result further confirms the conclusions drawn from the GridWorld experiments (Section 4.1.2) and positions TATC as a competitive alternative to LAP-REP in this difficult setting.

### 4.2.2 THE LEARNED SKILLS

We evaluate the exploratory potential of TATC's skills. Here, we compare the learned skills against 2 task-agnostic skill discovery methods, DIAYN (Eysenbach et al., 2019), and DCO (Jinnai et al., 2020). DIAYN learns a diverse set of skills by maximizing mutual information between skills and states. Similarly to our skills, DCO training is also based on a temporally-contrastive representation. In this case, DCO requires a pretrained LAP-REP (that approximates the Laplacian's second eigenvector). We train the required representation, as well as DCO, with the advantage of data collected from a *uniform* prior over $S$. For fairness, we train 8 skills for both methods (DCO and DIAYN).

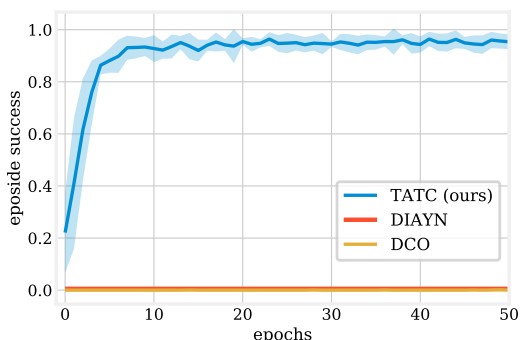

Figure 8: Skills Evaluation: performance gathered from 5 independent runs and then exponentially smoothed (0.9) for better visualization.

Once trained, the skills learned by each method are fixed and used to train a discrete high-level policy that can select across the available skills to solve a goal-reaching task with a sparse reward function $r_t = \mathbb{1}\left[\|s_{t+1} - g\|_2 \leq \epsilon\right]$. The sparsity of the reward naturally poses a challenge as no additional signal can guide the agent towards the goal, unlike the evaluation setting of DCO and DIAYN by Jinnai et al. (2020). The results (Figure 8) show that the skills learned by TATC quickly assist to complete the task while the skills learned with DCO and DIAYN do not. DIAYN's limited performance in difficult sparse-reward navigation tasks was also confirmed by Kamienny et al. (2021). These results suggest that in order to succeed, DCO

and DIYAN skills may require a richer signal like the distance-based dense reward used by Jinnai et al. (2020) to evaluate both of them – and where they show similar performances.

## 5 RELATED WORK

The potential of graph Laplacian representations in capturing functional information about the environment has motivated their use in the task-agnostic RL setting (e.g., Mahadevan, 2005; Machado et al., 2017; 2018). These are powerful tools that proved to scale to the continuous case (Wu et al., 2019; Machado et al., 2018; Jinnai et al., 2020). While these recent works proved to learn useful representations, they overlooked the challenging exploration problem that emerges when collecting the representation training data. Our framework proposes an alternative approach that explicitly couples the exploration challenge with the representation learning objective.

This work also relates to self-supervised learning (Bromley et al., 1994; Chopra et al., 2005), which brought recent advances in representation learning (Bachman et al., 2019; He et al., 2020; Chen et al., 2020; Grill et al., 2020; Caron et al., 2020). These techniques have naturally been adapted to RL, especially contrastive methods. While some of these benefited from visually contrasting observations (Laskin et al., 2020; Yarats et al., 2021), others leveraged temporal contrasts to learn representations (Mazoure et al., 2020; Stooke et al., 2021; Li et al., 2021), which fall closer to our work.

We designed our covering policy as a hierarchical agent. This has actually been the default setting to model temporally-extended actions (Sutton et al., 1999). Our work shares the same motivation as Vezhnevets et al. (2017) for training skills to follow latent directions. Among the large body of work on skill discovery, the eigenoptions framework (Machado et al., 2017) and its extensions (Machado et al., 2018; Jinnai et al., 2020) are probably the closest to our skill training scheme. Moreover, Eigenoptions also fit in the directional skills definition as they are trained to travel along the directions defined by the eigenvectors of the Laplacian (dimensionality of $|\mathcal{S}|$, potentially large). To contrast, we train directional skills defined by an arbitrarily diverse set of directions in the *learned* representation space (small dimensionality) to progressively learn about the environment. The incremental discovery paradigm has been previously adopted, either for exploration (Ecoffet et al., 2021), incremental skill discovery (Jinnai et al., 2020; Pong et al., 2019), or even state abstraction (Misra et al., 2020). Finally, we use learned skills to penalize boredom (Schmidhuber, 1991; Oudeyer et al., 2007; Oudeyer & Kaplan, 2009) in the representation space and encourage exploration. The idea of using skills to foster curiosity has also been investigated by Bougie & Ichise (2020).

## 6 CONCLUSION

The Laplacian representation as proposed by Wu et al. (2019) made the benefits of spectral methods affordable in large state spaces where function approximation is required. Unfortunately, this representation's quality is strongly tied to the uniformity of its training data distribution, as shown is Section 4. This has motivated the method proposed in this work where we reconcile similar temporally-contrastive representations with exploration demanding settings. Our approach leverages the practical skills' training that such representations allow, and uses the learned skills to better cover the state space and hence learn a better representation. In addition, we propose to augment the temporal contrast-based representation objective with temporal abstractions captured by the acquired skills. This has two benefits: it enforces the representation's dynamics-awareness, and contributes to exploring the environment by inducing boredom-fighting curiosity in the covering policy. We validate our method in tabular as well as continuous environments. Our representation learned in a non-uniform prior setting shows a comparable representational power to the one acquired from a uniform prior, and proves to scale well to challenging settings. Moreover, the skills that emerge from our algorithm show competitive performance in hard continuous control tasks with sparse rewards where standard skill discovery methods fail. Thus, with these results, we hope to bring such representations' applicability one step closer to realistic contexts.

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

# A    TEMPORAL ABSTRACTIONS-AUGMENTED TEMPORALLY-CONTRASTIVE LEARNING (TATC) IN THE NON-UNIFORM PRIOR SETTING

The proposed approach consists in a simultaneous training of the representation $\phi$ and the hierarchical agent $(\pi_{\text{low}}, \pi_{\text{hi}})$. The idea is to progressively extend the explored area while maintaining the previously collected knowledge. To do so, in the non-uniform prior setting, the agent switches with some probability $p_{rw}$ between following a uniformly random policy $\pi_\mu$ and executing the hierarchical policy (skills). The latter helps reach further areas, more efficiently, where data collected by $\pi_\mu$ would be used to train the representation $\phi$. Along their training, the skills would progressively extend to reach newly discovered areas, advancing the exploration frontier. Algorithm 1 provides a pseudocode of the proposed approach, in the non-uniform prior setting.

---

**Algorithm 1** TATC in the non-uniform prior setting

---

1: **Input:** $L, c, p_{rw}, N$
2: **for** $iteration = 1, 2, \ldots$ **do**
3:     $D_{\pi_\mu} = \emptyset, D_s = \emptyset$
4:     **for** $batch = 1, 2, \ldots, N$ **do**
5:         Reset to $s_0$ with probability $p_r$.
6:         $p \sim Unif([0,1])$
7:         **if** $p < p_{rw}$ **then**
8:             Run the uniformly random policy $\pi_\mu$ to collect $L$ random walk trajectories $\{\tau'_i\}_{i=1}^L$ of $c$ steps each.
9:             $D_{\pi_\mu} \leftarrow D_{\pi_\mu} \cup \{\tau'_k\}_{k=1}^L$
10:         **else**
11:             Run $(\pi_{\text{hi}}, \pi_{\text{low}})$ to collect $L$ consecutive skills' trajectories $\{(\tau_k, \boldsymbol{\delta}_k)\}_{k=1}^L$ and their corresponding directions
12:             $D_s \leftarrow D_s \cup \{(\tau_k, \boldsymbol{\delta}_k)\}_{k=1}^L$
13:         **end if**
14:     **end for**
15:     Optimize the policies $(\pi_{\text{hi}}, \pi_{\text{low}})$ using their intrinsic objectives 4 and 3 (vanilla actor-critic update)
16:     Optimize $\phi$ so as to minimize $\mathcal{L}_{\text{TATC}}(\phi; \mathcal{D}_s, \mathcal{D}_{\pi_\mu})$ (Eq. 6).
17: **end for**

---

# B    REPRESENTATION OBJECTIVE AUGMENTATION: ABLATION STUDY

## B.1    BOREDOM AUGMENTATION HELPS EXPLORATION

In order to illustrate the importance of the proposed augmentation – with the boredom term $\mathcal{B}$ – in the final objective (Eq. 6,) we conducted the same representation learning experiments for the three gridworld domains in the non-uniform prior setting, but this time with the non-augmented representation learning objective ($\beta' = 0$).

Figure 9 shows how the agent failed at exploring the whole domain. In T-MAZE, it focuses only on one corridor without getting curious about the other one. Regarding U-MAZE and 4-ROOMS, the agent stops exploring after discovering the end of the first corridor and the second room respectively. This is due to the lack of incentive to visit the yet unseen states, as they are less rewarding for $\pi_{\text{hi}}$ (i.e. closer in the representation space, hence smaller $R^{\text{hi}}$) than the furthest explored state. The effect of the proposed augmentation would compress the representation of the explored area, say the first corridor in U-MAZE, which makes the rest of the environment more appealing to explore for $\pi_{\text{hi}}$ (i.e. further in the representation space, hence larger $R^{\text{hi}}$). This emphasizes the importance of the boredom term in inducing the agent's exploratory behavior.

## B.2    BOREDOM AUGMENTATION ENFORCES DYNAMICS-AWARENESS

To verify the benefit of the boredom term beyond helping exploration, we train the representation with the non-augmented objective ($\beta' = 0$) but this time in the uniform prior setting, so that to

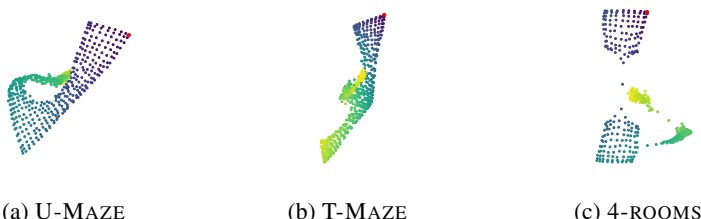

| (a) U-MAZE | (b) T-MAZE | (c) 4-ROOMS |

Figure 9: Learned representations in the gridworld domains with the *non-augmented* objective. Without the boredom term, the agent fails to cover the state space (cf. Figure 3), and may settle for incomplete representations. The colors reflect the distances in terms of the dynamics. They can be seen as quantities proportional to the length of the shortest path from the $s_0$ (marked in red) to the represented state.

marginalize the exploration problem. Figure 10 illustrates the learned representations in the three gridworld domains. These representations have failed to capture the dynamics. For example, in the case of 4-ROOMS, the distances from the first room to the fourth and third rooms are comparable in the representation space, which indicates that the representation does not take into account the relative order in which the rooms should be visited, when moving from the first room to the last. Similarly, in U-MAZE, the end of the maze is closer to the initial area than the second corner is. However, in order to reach the former on must pass by the latter. This proves that the boredom term is not only important for the desired exploratory behavior (cf. Figure 9), but also enhances the dynamics-awareness of our representation.

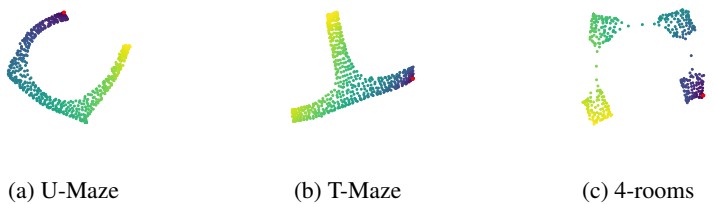

| (a) U-Maze | (b) T-Maze | (c) 4-rooms |

Figure 10: Learned representations when uniformly sampling over the state space. Without the boredom term, the representation does not reflect temporally-extended dynamics. The colors reflect the distances in terms of the dynamics. They can be seen as quantities proportional to the length of the shortest path from the $s_0$ (marked in red) to the represented state.

## C   IMPLEMENTATION DETAILS

### C.1   GRIDWORLD

For all the experiments, we defined the representation network as an MLP of two hidden layers of size 128 and tanh activations and a linear output layer of the size of representation's dimensionality $d$. The high-level and the low-level policies are both MLPs of two hidden layers of size 128 with tanh activations and a logsoftmax output layer of the size of their respective action spaces: the environment's 4 actions for the low-level policy and 8 actions for the high-level policy corresponding to the 8 directions $\Omega = \{(\cos(2k\pi/n), \sin(2k\pi/n)) \mid k \in \{0, ..., 7\}\}$ that define diverse skills.

The policies were trained with vanilla A2C with MC returns from the collected trajectories (Monte-Carlo estimates), i.e. no bootstrapped values where used. The skills being of a fixed size they could be trained without any reward discount ($\gamma = 1$). The high-level and low-level policies were entropy-regularized with the coefficients 0.3 and 0.1 respectively.

All of these networks were trained with RMSprop (Hinton et al., 2012) and a learning rate of 0.001. Environments specific hyperparameters are provided below.

### C.1.1 REPRESENTATION LEARNING

**U-MAZE.** Our representation is learned in the non-uniform prior setting with $p_r$=0.3, $p_{rw}$=0.4 and $K$=90 (around the number of steps between $s_0$ and the furthest state in the maze). We learn a 2-dimensional representation ($d = 2$) using the representation learning objective 6 with $\beta = 0.2$ and $\beta' = 2$. We fix the skills length to $c = 30$ steps (so $L = K/c = 3$), and jointly train the representation $\phi$ and the policies ($\pi_{\text{hi}}, \pi_{\text{low}}$) by collecting, for each update, a batch of $N = 32$ trajectories of length $c$ to fill $D_s$ and $D_{\pi_\mu}$ as described in Algorithm 1. We train them for 700 epochs where each epoch corresponds to 10 updates (convergence to the complete representation required around 500 epochs).

**T-MAZE.** Our representation is learned in the non-uniform prior setting with $p_r$=0.2, $p_{rw}$=0.4 and $K$=40 (around the number of steps between $s_0$ and the furthest state in the maze). We learn a 2-dimensional representation ($d = 2$) using the representation learning objective 6 with $\beta = 0.2$ and $\beta' = 2$. We fix the skills' length to $c = 20$ steps (so $L = K/c = 2$). and jointly train the representation $\phi$ and the policies ($\pi_{\text{hi}}, \pi_{\text{low}}$) by collecting, for each update, a batch of $N = 48$ trajectories of length $c$ to fill $D_s$ and $D_{\pi_\mu}$ as described in Algorithm 1. We train them for 700 epochs where each epoch corresponds to 10 updates (convergence to the complete representation required around 350 epochs).

**4-ROOMS.** Our representation is learned in the non-uniform prior setting with $p_r$=0.25, $p_{rw}$=0.5 and $K$=60 (around the number of steps between $s_0$ and the furthest state in the maze). We learn a 2-dimensional representation ($d = 2$) using the representation learning objective 6 with $\beta = 0.2$ and $\beta' = 2$. We fix the skills' length to $c = 20$ steps (so $L = K/c = 3$). and jointly train the representation $\phi$ and the policies ($\pi_{\text{hi}}, \pi_{\text{low}}$) by collecting, for each update, a batch of $N = 32$ trajectories of length $c$ to fill $D_s$ and $D_{\pi_\mu}$ as described in Algorithm 1. We train them for 700 epochs where each epoch corresponds to 10 updates (convergence to the complete representation required around 350 epochs).

The Laplacian representation (LAP-REP) was trained in the same environments' settings described above, for both the uniform and non-uniform prior settings (of course no policy is trained here so $p_{rw} = 1$, and ($s_0, p_r$) are not relevant for the uniform prior setting). We used the representation learning objective and the associated hyperparameters proposed by Wu et al. (2019). For the uniform prior setting, our online data collection does not cause any discrepancy compared to the offline scheme used in Wu et al. (2019). Indeed, for a minibatch size large enough, the stochastic minibatch based training of LAP-REP when using a **uniform** prior is agnostic to the data collection sheme (offline vs online) since in both cases the minibatches are sampled from the exact same uniform distribution over the state space.

### C.1.2 PREDICTION AND CONTROL

In the prediction and control experiments, we evaluate each pretrained representation by training an actor-critic agent to solve a goal-achieving task with a sparse reward ($r = 1$ upon reaching the goal). The episode size was set to 100 steps for all the gridworld domains.

For the prediction, the critic head is a linear function in the given representation, while the actor is a MLP with two hidden layers of size 64 and tanh activations, a logsoftmax output layer of size 4 (discrete gridworld actions) and the actor's input is the state one-hot code. For the control experiments, the actor-critic agent is defined on top of the representation as a MLP of two hidden layers of size 64 with tanh activations that feed two output heads: a linear critic head and a logsoftmax action head for the 4 actions. The agent is trained with A2C with MC returns and a discount of $\gamma = 0.98$, a batchsize of 80 episodes, an entropy regularization with a 0.01 coefficient and Adam optimizer (Kingma & Ba, 2014) with a learning rate of 0.001.

### C.2 MUJOCO: ANTMAZE

In this navigation task, the environment is composed of $4 \times 4 \times 4$ blocks defining a U-shaped corridor. The environment's action space is 8 dimensional. For the sake of simplifying the RL training algorithm, we mapped each dimension values interval to a discrete set of 5 values equally spaced over the interval. We used the same architectures for the representation and the policies as

for the gridworld, with the only difference that for the low-level policy, the action head was adapted to the discretization of the action space by having 8 logsoftmax output heads of size 5, one for each action dimension and the corresponding 5 discrete values. This choice makes the training algorithm simpler as it allows using A2C here as well.

Our representation is learned in the non-uniform prior setting with $p_r = 0.2$, $p_{rw} = 0.3$ and $K = 500$. We learn a 2-dimensional representation ($d = 2$) using the representation learning objective 6 with $\beta = 0.2$ and $\beta' = 5$. We fixed their length to $c = 100$ steps (so $L = K/c = 5$). and jointly train the representation $\phi$ and the policies $(\pi_{\text{hi}}, \pi_{\text{low}})$ by collecting, for each update, a batch of $N = 32$ trajectories of length $c$ to fill $D_s$ and $D_{\pi_\mu}$ as described in Algorithm 1. We train them for 1000 epochs where each epoch corresponds to 10 updates (convergence to the complete representation required around 650 epochs).

The policies were trained with the same A2C used in gridworld domains and the same RMSprop hyperparameters. The high-level and low-level policies were entropy-regularized with the coefficients 0.15 and 0.1 respectively.

### C.2.1 Reward Shaping

Regarding the Laplacian representation baseline, LAP-REP was learned in the same non-uniform prior setting described above, with the representation objective and its associated hyperparameters proposed by Wu et al. (2019). In this setting, the data collection and the representation training are performed simultaneously in an online fashion. We have also tested the offline representation training, replicating the training scheme in Wu et al. (2019). Still in the non-uniform prior setting, we collected 500000 training samples (10 times more than in Wu et al. (2019)) according to a uniformly random policy, then we trained the representation on the large dataset built this way. For all other hyperparameters, we used the same as provided in Wu et al. (2019). Both trainings ended up giving the same performance for the reward shaping task.

Now, for the reward shaping, we train a Soft Actor-Critic (SAC) (Haarnoja et al., 2018) agent to reach a goal area (neighbourhood around the goal state) with episodes of size 1000 steps. We use the following hyperparameters:

- Discount $\gamma = 0.99$
- Entropy coefficient (temperature) $\alpha = 0.1$
- Soft critic updates with smoothing constant $\tau = 0.005$
- Replay buffer of size $5 \cdot 10^6$ (equal to the number of training steps).
- Adam optimize with learning rate of 0.0001

As SAC is sensitive to the reward scale (Haarnoja et al., 2018), we grid-searched this hyperparameter in $\{10^{-5}, 10^{-4}, \cdots, 1.0, 2.0\}$, and the best performing one for our representation was 1.0, while for LAP-REP the SAC agent didn't succeed with any of these values to solve the task.

### C.2.2 Skills Evaluation

To train DCO, we first collect a dataset to estimate the second eigenvector and then use the same dataset to train a policy – the option – using DDPG (Lillicrap et al., 2015). Each DCO option is tied to its own eigenvector estimate and its own training set of size 500000 (10 times the size used in Jinnai et al. (2020)). As suggested by the authors of DCO (Jinnai et al., 2020), the remaining hyperparameters to estimate the eigenvectors and train their corresponding options were taken from Wu et al. (2019). DIAYN skills were trained as recommended by Eysenbach et al. (2019). For fair comparison, we train 8 skills for both DCO and DIAYN.

For the skills evaluation stage, we freeze the learned low-level policies and train a high-level policy to use the 8 skills as the only available actions to reach the goal $g$ on the other end of the AntMaze environment using a **sparse** reward $r_t = \mathbb{1}\left[\|s_{t+1} - g\|_2 \leq \epsilon\right]$ within a finite horizon of 1000 steps . Note that this tasks is quite challenging given the type of reward and the length of episode especially in a continuous state space. As our skills offer some flexibility in their execution (can be started everywhere and run for arbitrary number of steps), this episode length was decomposed to 5 skills

of 200 steps each. The high-level policy was trained with A2C with MC returns (no discount given the finite horizon) a batch size of $8$ episodes, and RMSprop optimizer with a learning rate of $0.001$.

## D   THE SWITCHING UTILITY OF THE BOREDOM TERM

Note that $\mathcal{D}_s$, in Eq. 5, may contain trajectories from skills that are not yet duly trained; for example early in the training or in a freshly discovered area. Since at that stage, these skills' trajectories are close to random walks, their contribution in the boredom term is similar to the first attractive term, in Eq. 2, which is based on random walks. This means that a new skill trajectory initially contributes to the temporal similarity term (attractive term) in training the representation, thus making the most out of the sampled skills' trajectories while these are still early in their training. The more a skill is trained, the more structured its trajectories become and the more they contribute to the intended "boredom" effect (Section 3.3), that is encouraging exploration and dynamics awareness (Appendix B).

