# OpenReview forum: "Temporal abstractions-augmented temporally contrastive learning: an alternative to the Laplacian in RL"
_ICLR.cc/2022/Conference — ICLR 2022 Submitted_

### Official Review · Reviewer_XcUD · 2021-10-22

**Correctness:** 3
**Technical Novelty And Significance:** 3
**Empirical Novelty And Significance:** 2
**Recommendation:** 5
**Confidence:** 4

**Main Review:**

The paper's topic is not really specific to Laplacian RL but proposes an exploration method which seems specific to the exmployed evaluation environments of Laplican RL. As Laplacian RL only works on state spaces with fully reversible actions to allow modelling the state space as an undirected graph, the method has only been evaluated on relatively somply deterministic 2D routing tasks. In general, one could generally argue how practicall important fully reversible environments are as they basically do not yield probabilistic interesting decisions.
The proposed method for  using directions to push exploration away from the known part of the state space, does not really seem task-agnostic. Furthermore, the contributions should have been discussed in the light of other methods that improve exploration based on intrinsic rewards. An open question is whether approach could be extended to more general environments where the definition of directions is less obvious.
Admittedly, the two compared task-agnostic skill discovery methods are clearly beaten on the task. However, given that the proposed exploration scheme does not seem to be straight forward to extend to more complicated environments that is not too suprising.

---- Post author response----
Though the authors claim that exploration is not the main goal of the proposed approach, it is not clear what the difference between a task agnostic exploration method like curiosity with intrinsic and the proposed skill-based covering policy is.
After reading author response I reviewed the description of the  AntMaze and have to admit that in this setting the state space seems considerably larger as the ant's joints need to be moved.  However, I did not find a proper description of the environment as the corresponding reference is MuCojo and the description in the paper and the appendix do not really offer all details on the environment.




**Summary Of The Paper:**

The paper proposes an exploration strategy for adapting Lapliacan RL to settings without the possibility to uniformely sample from the state space. In particular, the work builds on the Wu et al. discussion of estimating a spectral decomposition of the state space. However, the estimation of the Laplacian's eigenvectors is based on uniform sampling from  states and possible transitions.
To address this restriciton the authors propose a hierarchical exploration scheme, which follows directions on a higher level and optimizes actions of the lower level. Based on these levels the authors propose to employ intrinsic rewards to improve exploration and achieve a better cover of the state space. Experiments are conducted on grid graphs and continous navigation tasks and demonstrate that the new sampling policy performs better than sampling from random walks. More interestingly the authors compare to two other task-agnostic skill discovery
methods.


**Summary Of The Review:**

The contribution of the paper does not seem to be strongly connected to Laplacian RL but seems to propose an hierarchical temporal abstraction method for improving exploration. The result that the approximation scheme of Wu et. al does not work well if the sampling is biased around the starting state and thus, non-uniform is not too surprising.

Thus, the paper should discuss and compare its contribution to other exploration schemes based on intrinsic rewards.
Furthermore, the authors should address the limitations of their approach w.r.t. the properties of the environment.

---

### Official Review · Reviewer_2jVQ · 2021-10-31

**Correctness:** 3
**Technical Novelty And Significance:** 3
**Empirical Novelty And Significance:** 2
**Recommendation:** 6
**Confidence:** 4

**Main Review:**

**Strengths:**

* The requirement of a uniform prior in (Wu et al., 2019) is indeed a big limitation. Overcoming this limitation is an interesting topic and will extend the applicability of Laplacian representation.
* The writing is good and clear. Most notations are well defined. The figures are also informative.
* Most experiments are well executed.


**Weaknesses:**
* One of my concerns about this paper is the lack of theoretical support. After reading the first half of the abstract, I was very excited and expecting some provable results that overcome the uniform limitation (or say something about the inevitability of the uniform prior, which would also be interesting). Unfortunately, all evidences are empirical. I am not saying that TATC does not work well. TATC is effective, as demonstrated in the experiments. It is just that I feel without theoretic support, the paper is a little bit weak.
* Also, I feel that the motivation and the conclusion are somewhat mismatched. The motivation is that approximating Laplacian representation (Wu et al., 2019) suffers the limitation of requiring a uniform prior. The conclusion is that TATC can learn a better representation without access to a uniform prior. However, the learned representation by TATC may (probably) NOT be the Laplacian representation. I think it would be better to say "Motivated by Laplacian representation, we propose a new temporal contrastive representation learning framework in RL" rather than "We overcome the uniform prior limitation in (Wu et al., 2019)".
* Quite a few details about the experiments are missing. I will explain below.

**Other comments:**

* What is the state (or observation) space for AntMaze environment?
* How are the colors in Figure 3 and Figure 6 computed? My guess is that they are not calculated in the representation space but manually specified instead.
* For the prediction and control experiments (Section C.1.2), is the starting state fixed at a specific state $s_0$ across episodes? Where are the goals located?
* How does the low-level policy network take in the direction vector $\delta$ ?
* I am wondering how the hyper-parameters in Section C.1.1 and C.2 (e.g., $p_r$,  $p_{rw}$, $\beta'$) are chosen? Are the results robust to different hyper-parameter choices?
* In this paper, the authors use $|\Omega|=8$ unit vectors as the representative directions in a 2-dimensional representation space. I am wondering how to deal with higher dimensions, since the number of representative directions grows exponentially with the dimension of the representation space.
* In Section 4.1, the authors emphasize that "no positional information is provided to the agent". I do not see the particular necessity of hiding position information. Even with the (x, y)-position, it is still challenging to learn a dynamic-aware representation, due to the presence of walls.
* In Section 4.1.1, why the AntMaze environment used in this paper is larger compared to the one used in (Wu et al., 2019)? I feel the AntMaze here is pretty similar to the ant-maze2 in (Wu et al., 2019).
* For the experiments in Figure 7, can you add comparisons with Lap-Rep (trained with total number of dimensions d=2)? Though it still suffers from the uniform prior limitation, I believe Lap-Rep (d=2) can achieve comparable performance as TATC.
* I am not quite convinced by the results in Section B.2. This setting is similar to the original Lap-Rep paper, except that the implementation of the repulsive term is different. Therefore, I would expect similar results as Lap-Rep. For Lap-Rep (here d=2), the learned representation would span the same subspace as 2 smallest eigenvectors. Since the smallest eigenvector is constant, the distance between two states would then depend on the 2nd smallest eigenvector (the Fiedler vector), and accordingly reflect the environment dynamic (at least for a chain-like environment). However, for U-Maze and 4-Room, it seems that the learned representations in Figure 10 do not have this property. For example, the distance between two ends of the U-Maze is smaller. I would like to hear opinions from the authors.
* Section 2.1: It might be clearer to give a definition of $\Delta$.

* Please consider including following two related works:
  * Wang, K., Zhou, K., Zhang, Q., Shao, J., Hooi, B., & Feng, J. (2021, July). Towards Better Laplacian Representation in Reinforcement Learning with Generalized Graph Drawing. In *International Conference on Machine Learning* (pp. 11003-11012). PMLR.
  * Machado, M. C., Barreto, A., & Precup, D. (2021). Temporal Abstraction in Reinforcement Learning with the Successor Representation. *arXiv preprint arXiv:2110.05740*.

**Summary Of The Paper:**

This paper introduces a representation learning framework for reward-agnostic RL, which is based on temporal contrastive learning. The proposed method (TATC) is motivated by the Laplacian representation approximation approach (Wu et al., 2019) and overcomes the limitation of requiring a uniform prior. The experiments demonstrate the effectiveness of TATC.

**Summary Of The Review:**

Overall, I do not think this paper addresses the uniform prior limitation of Lap-Rep. It is an ok paper if it positions itself in proposing a new representation learning framework. If so, experiments on more challenging environments are necessary. Therefore, I lean towards rejection.

---

### Official Review · Reviewer_73Mt · 2021-11-06

**Correctness:** 3
**Technical Novelty And Significance:** 2
**Empirical Novelty And Significance:** 2
**Recommendation:** 5
**Confidence:** 4

**Main Review:**

Strengths:
- Significance: This work is related to the exploration issue, which is an important topic in RL and will be intriguing to the community.
- Soundness: The proposed method is sound to me.


Weakness:
- The exhibition of the paper should be improved and some aspects are hard to follow. For example, there are too many overlapping contents in the section of the introduction, related work, and conclusion. However, the presentation on the proposed method is unclear, such as the problem formulation, method design, and how it works in the procedure.
- Limited novelty: this work extends the previous work of LAP-REP (Wu et al., 2019) and it is incremental.
- Empirical evaluation is weak. The paper claims that the proposed method can scale better to challenging tasks, and the learned skills can solve difficult tasks.
However, the current results don't match its claims. I didn't see the results for the challenging or difficult tasks (e.g., atari games), although gridworld and AntMaze are good testbeds in demonstrating the efficiency of exploration.


Minors:
- Is there a typo in this sentence? "hence the need for an exploration strategy for a better covering distribution".
- Figure 3: it's better to give more detailed descriptions, such as what's the meaning of the shape and the different colors. Besides, how (a) can be evolved to (j), how does this work?

**Summary Of The Paper:**

This work focuses on the problem of uniformity in representation learning, especially Laplacian representation. To address the issue, the authors propose TATC method, which leverages the skills training that representations allow, and uses the learned skills to better cover the state space, i.e., learn a covering policy. Finally, the proposed method is evaluated on (discrete) gridworld and (continuous) navigation environments, with results and analysis demonstrating its effectiveness.

**Summary Of The Review:**

I prefer to weak reject (marginally below the acceptance threshold) this paper due to:
- the work is incremental.
- the empirical evaluation is weak.

---

### Decision · Program_Chairs · 2022-01-20

**Decision:**

Reject

**Comment:**

The topic of this paper is non-uniform priors and exploration in reinforcement learning with the graph Laplacian.

All reviewers appreciated several aspects of this work but they all also have several reservations.

Looking at the paper, reviews and discussions, I see the potential a very nice more general contribution. This potential is not fully realised as the paper stands now. Acceptance can therefore not be recommended.